# Is the Label Trustful: Training Better Deep Learning Model via Uncertainty Mining Net

## Abstract

In this work, we consider a new problem of training deep neural network on partially labeled data with label noise. As far as we know, there have been very few efforts to tackle such problems. We present a novel end-to-end deep generative pipeline for improving classifier performance when dealing with such data problems. We call it Uncertainty Mining Net (UMN). During the training stage, we utilize all the available data (labeled and unlabeled) to train the classifier via a semi-supervised generative framework. During training, UMN estimates the uncertainly of the labels' to focus on clean data for learning. More precisely, UMN applies the sample-wise label uncertainty estimation scheme. Extensive experiments and comparisons against state-of-the-art methods on several popular benchmark datasets demonstrate that UMN can reduce the effects of label noise and significantly improve classifier performance.

## 1 Introduction

Deep Learning (DL), to learn powerful representations, it usually requires a large amount of training data. However, for many real world problems, it is not always possible to obtain sufficiently large training data. What we can usually get is limited training data with corrupted labels which heavily affect the model performance. Although acquiring large data is not hard, considering the information explosion on the internet, accurate labeling is usually an expensive and error-prone task which involves humans' interaction, especially experts with knowledge in the specific field. Most of the time, we have to build DL models using limited training data with corrupted data labels. It becomes very challenging to apply current popular deep learning frameworks to solve this problem.

Training a deep learning model via noisily labeled data is a challenging task (Patrini et al., 2017; Li et al., 2017; Ding et al., 2018; Menon et al., 2015; Goldberger & Ben-Reuven, 2016; Jiang et al., 2017). Most of current explorations for dealing with such problems usually require large amount of labeled data as a prerequisite. It means that in these works, the designed pipelines and evaluations are based on having enough labeled training data, though they may include label noise. These approaches will suffer from the limited size of labeled training data as well as the label noise. As we know, the hierarchical representation learned by deep learning models mainly benefits from the amount of data. The deep learning model can easily overfit to the mislabeled samples especially when the data size is small (Tarvainen & Valpola, 2017; Ren et al., 2018).

Unfortunately, there is no general solutions to solve the problem of training on limited annotated data with label noise. If we directly train the deep learning model only with the labeled data, the small portion of labeled training data will limit the learning capability. Meanwhile, the label noise makes it very challenging to utilize the labeled data directly. In this work, our framework can handle these problems simultaneously. UMN includes two major components. The first part is used to learn a latent feature representation via a generative framework (VAE) using all the training data. Then the learned embedding is applied as the input to its subsequent semi-supervised learning component by conditional variational autoencoder to train the classifier. To handle the label noise, we integrate label uncertainty estimation module where the reliable data contributes more to the model training and noisy data contributes less.

Though recent works (Hataya & Nakayama, 2019; Ren et al., 2018) discuss such problems, they usually assume to have a separate small clean dataset to begin with. It may not be possible to have an independent clean dataset in real scenarios. Current supervised robust learning approaches cannot

be directly adopted to our case since the labeled data is small and not reliable. The limited labeled training data makes the supervised model overfit very easily to the small size of noisy data. This will heavily hurt the model performance. The limited data labels and noise can also affect other related state-of-the-art semi-supervised learning approaches. We will present the comparisons with these approaches in the experiment section.

In this paper, our major contributions can be summarized into the following aspects: First, our studied problem is very general in real scenarios: small mislabeled datasets with large amount of unlabeled data, and there are very few existing approaches to target this kind of problems. Second, instead of treating the label noise as the preknowledge, we explicitly use the deep generative architecture to model the noisy labels at the sample level precisely. This has been validated by the improved experimental results as well as the theoretical analysis. Third, we show how to use the moving average model to estimate the sample-wise uncertainty in labels. We have explained it in the theory analysis section and validated this assumption in the experiments. Furthermore, our experiments demonstrate that UMN can also help identify these mis-labeled data without any prior knowledge. Finally, we build one end-to-end deep learning pipeline to train against the noisy labeled data.

## 2    RELATED WORKS

Along with the popularity of deep neural network, the problem of training deep neural networks with noisy label data has started to draw our attention. Natarajan et al. (Natarajan et al., 2013) propose the unbiased estimator of the surrogate loss function and calculate theoretical bounds for empirical risk optimization. Modeling the consistency as the regularization for deep neural network is another route for robust learning with noisy labeled data. Reed et al. (Reed et al., 2015) design a robust loss to model the prediction consistency. In (Li et al., 2017), the authors propose a knowledge distillation framework where they train an auxiliary model on a small set of clean data samples and linearly combine its predictions with the observed labels to form the new targets to train. In (Ding et al., 2018), a two-stage approach is proposed. First a DNN model is trained on the noisy data and used to filter out these noisy samples. Then another model is trained on the filtered dataset using a semi-supervised approach by treating these filtered out samples as unlabeled. Menon et al. model the corruption process of a dataset by learning the class-probability estimator (Menon et al., 2015). Another way of regularization for training on noisy labels is to estimate the class conditional corruption ratio. Goldberger et al. (Goldberger & Ben-Reuven, 2016) propose a softmax layer along with the classifier to predict the class conditional corruption ratio. In (Jiang et al., 2017), they propose MentorNet which learns a data driven dynamic curriculum to be followed by the StudentNet.

Semi-supervised learning with few labeled samples is also a well studied area. Some of the closely related works are (Gordon & Hernández-Lobato, 2017), (Kingma et al., 2014), (Tarvainen & Valpola, 2017). Kingma et al (Kingma et al., 2014) propose a stacked deep generative semi-supervised model for training on partially labeled datasets. They first train a variational autoencoder on the labeled data, then stack on top of it a conditional variational autoencoder and continue training in semi-supervised fashion to get a robust classifier. In (Tarvainen & Valpola, 2017), the authors train a classifier in semi-supervised way using temporal ensemble approach. They train a student network and maintain a teacher network as the exponential moving average of the student. They use classification loss for labeled data and enforce consistency loss between teacher and student for unlabeled data.

There have been limited works dealing with bi-quality data (Hataya & Nakayama, 2019), (Langevin et al., 2018). Bi-quality data is defined in (Hataya & Nakayama, 2019) as datasets with few labeled samples where the labels are potentially corrupted. In (Langevin et al., 2018), the authors discuss to use the deep generative semi-supervised model to model noisy labels. They assume that the labels are also randomly corrupted with a probability $\rho$. Their idea relies on the predefined $\rho$ to train the classifier. However, the corruption probability $\rho$ is not known as a prior in reality.

Compared to these work, we present a new architecture for training a robust classifier with dataset including noisy labels. We assume that there is no preknowledge of the label corruption rate and we do not have access to a separate clean dataset which is more close to the real scenario.

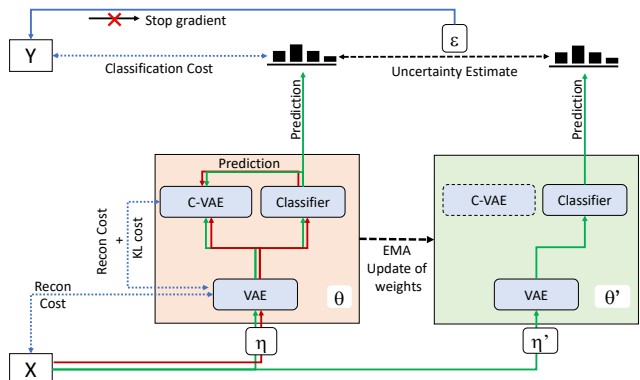

Figure 1: An overview of UMN framework: $X$ and $Y$ are the observed inputs and potentially corrupt labels. $\theta$ and $\theta'$ are semi-supervised generative models. $\eta$ and $\eta'$ are noise functions that perturb the input $X$. The red and green colored arrows represent the flow of unlabeled and labeled data through the models. $\theta$ is updated via a stochastic gradient descent approach so as to minimize the classification and VAE losses, and $\theta'$ is the Exponential Moving Average (EMA) of $\theta$. The sample-wise uncertainty ($\epsilon$) between the predictions of $\theta$ and $\theta'$ are used to re-weigh the gradients that are back propagated to $\theta$ via the classification loss. The general architecture for $\theta$ is adapted from (Kingma et al., 2014). VAE indicates Variational AutoEncoder and C-VAE represents Conditional-VAE. Note that in $\theta'$, when making predictions (on labeled data) only the weights from encoder of the VAE and the classifier are used. For simplicity, the KL loss for the predictions on unlabeled data with respect to an uniform prior is not shown. For simplicity, we omit the optional consistency loss term between the classifiers of $\theta$ and $\theta'$ for unlabeled data in the figure.

## 3 METHOD

The goal of our work is to learn a better model with the limited labeled data including the noise. Since the labeled data only occupies a small portion of the training data, the labeled data alone is not sufficient to train a good model. At the same time, we also need to solve the label noise problem. To deal with these issues, our pipeline consists of two major components. In the first part, we apply unsupervised learning scheme to learn a latent feature representation via all the available training data. Then we utilize the label via semi-supervised learning pipeline to train the classifier. At the same time, we incorporate the label uncertainty estimation to reduce the influences coming from these mislabeled data. We estimate the uncertainty of the given sample from the prediction of the updating model and its guider model (exponential moving average of the updating model), part of this idea is inspired by work (Polyak & Juditsky, 1992). The estimated uncertainty is then used to assign weights to the training data samples. This is an end-to-end framework. We call it Uncertainty Mining Nets(UMN). In the following subsection, we will introduce UMN in more details.

### 3.1 LATENT-FEATURE LEARNING

In this work, we represent the given data as the set of $(X, Y) = \{(x_1, y_1), \ldots, (x_N, y_N)\}$ where $x_i$ is the observed data sample and $y_i$ is its corresponding label which may be potentially corrupted. If $x_i$ is unlabeled, then $y_i$ becomes empty. To be concise, we omit the index $i$ in the rest of the paper.

Motivated by the recent success of generative models in semi-supervised learning related applications, firstly, we apply the variational autoencoder (VAE) on all the training data. VAE consists of two modules: an Encoder that maps the variables $x$ to the latent variables $z$ to approximate the prior distribution of $p(z)$ and a Generator $p_\theta(x|z)$ that samples the the input variables $x$ given the latent variables $z$. It can be formulated as

$$p(z) = \mathcal{N}(z|0, I), p_\theta(x|z) = f(x; z, \theta); \tag{1}$$

where $f(x; z, \theta)$ is the likelihood function and $\mathcal{N}$ indicates the Gaussian distribution. $\theta s$ are network parameters. The goal of VAE is to maximize the evidence variational lower bound (ELBO) of $p_\theta(x)$ as $\log p_\theta(x) \geqslant \mathbb{E}_{q_\phi(z|x)} \log p_\theta(x|z) - D_{KL}(q_\phi(z|x)||p(x))$.

The learned latent variables obtained from VAE can be used as the feature representation to train the classifier. Based on this, we can fully utilize the knowledge from the training data without the constraint of the labeled data size limitation. The low-dimensional embedding $z$ can well cover the distribution of the input data $x$. Our hypothesis is that the unlabeled data helps training robustly against the noisy labels via the deep generative modeling. We will see this in the next subsection.

## 3.2 SEMI-SUPERVISED LEARNING

After building the unsupervised learning framework via VAE, we apply the label information to improve the learning capability of the neural network. Instead of using the raw data $x$, we directly use the latent vector $z_a$ obtained from VAE of latent-feature learning part along with the pre-defined labels. We follow the idea of conditional generative model.

Our generative process can be formulated as,

$$y, z_b \sim M(y),\ p(z_b), \quad z_a \sim p_\theta(z_a|z_b, y), \tag{2}$$
$$x \sim p_\theta(x|z_a), \quad \hat{y} \sim p_\theta(\hat{y}\,|\,y, z_a); \tag{3}$$

where $M$ is the multinomial distribution. $y$ indicates the true label. $z_a$ is jointly generated from $z_b$ and $y$ and $z_b$ is the latent representation of the input. In this work, we treat $y$ as the latent variables. We use $\hat{y}$ to denote the observed labels. To more accurately capture the reliability of labels, we estimate each labeled sample uncertainty via $p_\theta(\hat{y}\,|\,y, z_a)$. the posterior distribution can be factorised as

$$q(z_a, z_b, y|x, \hat{y}) = q(z_a|x)\, q(y|z_a)\, q(z_b|z_a, y). \tag{4}$$

Following this, the Evidence Lower Bound (ELBO) can be derived as follows:

$$- \sum_{x \in \{L, U\}} \mathbb{E}_{q_\phi(z_a|x)} \sum_{y \in C} q_\phi(y|z_a) \mathrm{KL}(q_\phi(z_b|z_a, y) \parallel p(z_b))$$

$$- \sum_{x \in \{L, U\}} \mathbb{E}_{q_\phi(z_a|x)} \sum_{y \in C} q_\phi(y|z_a) \mathrm{KL}(q_\phi(y|z_a) \parallel p(y))$$

$$- \sum_{x \in \{L, U\}} \mathbb{E}_{q_\phi(z_a|x)} \sum_{y \in C} q_\phi(y|z_a) \mathbb{E}_{q_\phi(z_b|z_a, y)} \big( \log p_\theta(z_a|z_b, y) - \log q_\phi(z_a|x) \big)$$

$$+ \sum_{x \in \{L, U\}} \mathbb{E}_{q_\phi(z_a|x)} \log p_\theta(x|z_a) + \sum_{x \in \{L\}} \mathbb{E}_{q_\phi(z_a|x)} \sum_{y_k \in C} q_\phi(y_k|z_a) \log p_\theta(\hat{y}|y_k, z_a), \tag{5}$$

where $q_\phi(y|z_a), q_\phi(z_a|x), q_\phi(z_b|z_a, y), p_\theta(z_a|z_b, y), p_\theta(x|z_a)$ indicate the classifier, encoder, conditional encoder, conditional decoder and decoder functions respectively. $L$ indicates labeled data. $U$ means unlabeled data and $C$ denotes the set of used classes. The last term can be treated as labeled loss term as the supervised portion of Eq. 5.

For a more detailed derivation of the ELBO, please refer to the Appendix. As indicated in Eq. 5, except the last term, all other terms in the equation are calculated over all training data samples. The last term serves as the supervised factor for semi-supervised learning where the observed labels may be incorrect. It uses an uncertainty related function $p_\theta(\hat{y}|y_k, z_a)$ to modulate the sample weights during the gradient back-propagation. As indicated in Fig. 1, $p_\theta(\hat{y}|y_k, z_a)$ is approximated based on the sample-wise uncertainty $\epsilon$. More details are given in the following.

For simplicity, we can also represent the generative process of the observed $\hat{y}$ by Fig. 2.

One plausible way to estimate the uncertainty of the sample label is to model the class conditional corruption ratio $p(\hat{y}|y)$, which is a mean field approximation. As discussed in (Langevin et al., 2018), the class conditional probability can be computed as

$$p(\hat{y}\,|\,y) = \begin{cases} 1 - \epsilon, & \text{if } \hat{y} = y \\ \epsilon/(C-1), & \text{otherwise} \end{cases} \tag{6}$$

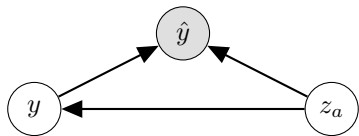

Figure 2: The graphical model generating the observed label

and the labeled loss term can be formulated as

$$\mathbb{E}_{q_\phi(z_a|x)}\big[f(\epsilon)q_\phi(y|z_a)\big]. \tag{7}$$

where $f(\epsilon) = \log[\frac{(C-1)(1-\epsilon)}{\epsilon}]$ is a function with constant $\epsilon$. From Eq. 7, we can see that it assigns the same weight for all samples belonging to the same category, i.e., all labeled terms contribute equally to the gradient back propagated regardless of their label correctness as

$$\mathbb{E}_{q_\phi(z_a|x)}\big[f(\epsilon)\nabla_\phi q_\phi(y|z_a)\big]. \tag{8}$$

Although this idea is simple and easy to follow, the assumption may not be the case in the real scenario. In general, the value of $\epsilon$ which is the class conditional corruption rate, remains unknown beforehand. Another issue is that it's not accurate to apply the same categorical corruption ratio to all data samples of a given class. Our approach solves these problems by estimating sample-wise uncertainty during the training stage.

We replace the class conditional $\epsilon$ in Eq. 9 with a point-wise estimation of the label uncertainty $\epsilon(z_a)$, where we follow the generative process of Eq. 3. The uncertainty of label is hence $\epsilon(z_a)$. Pointwise label uncertainty is calculated as

$$p(\hat{y} \,|\, y, z_a) = \begin{cases} 1 - \epsilon(z_a), & \text{if } \hat{y} = y \\ \epsilon(z_a)/(C-1). & \text{otherwise} \end{cases} \tag{9}$$

So $f(\epsilon)$ in Eq. 7 becomes $f(\epsilon) = \log[\frac{(C-1)(1-\epsilon(z_a))}{\epsilon(z_a)}]$. That is, samples contribute relatively more to the gradient backpropagation when the estimated $\epsilon$ is small which indicates there is a high probability that the given label is correct. By focusing on these reliable targets during training, we can train a more robust classifier. From this, we can see that the key is to approximate $\epsilon(z_a)$ precisely. In the following subsection, we discuss the proposed approach to model $\epsilon(z_a)$ using the differences in predictions from the updating classifier model and its moving average model.

We want to emphasize that although UMN is based on VAE framework, its architecture can be flexible. Depending on the task, the encoder of its generative model-VAE can be implemented using various deep learning networks, e.g., AlexNet, GoogLeNet, ResNet, etc. Meanwhile, UMN is not limited to one specific task, like image classification. It can be easily extended to other classification problems.

### 3.2.1 UNCERTAINTY ESTIMATION

As discussed above, we aim to train a more robust classifier in an end-to-end semi-supervised architecture. The key to this idea is to estimate sample-wise label noise ($\epsilon(z_a)$) as this cannot be known as a priori. We are motivated by the initial work of the iterate average model (Polyak & Juditsky, 1992) where exponential moving average of the model weights can be used to stabilize the training in stochastic gradient based approaches. Exponential moving average of the model weights can be calculated as:

$$\theta'_t = \gamma\theta'_{t-1} + (1-\gamma)\theta_t, \tag{10}$$

where $\theta$ is the set of weight parameters of the classifier model and $\theta'$ is its exponential moving average. $\gamma$ controls the smoothness of model updates. We name them $Learner$ and $Guider$ models respectively. $t$ is the step index of the iterative optimization. Since this iterate average gives optimal bound for convergence rate (Polyak & Juditsky, 1992) and can be less sensitive to the noisy updates (Eq. 10), we adopt it as the $Guider$ model to estimate the label uncertainty.

We approximate the label uncertainty via the absolute difference in the predicted probability of the $Learner$ and the $Guider$ for the observed class as following

$$\epsilon(z_a) = |\,f'(z_a) - f(z_a)\,|, \tag{11}$$

where $f'$ and $f$ indicate the classifier's predictions of the learner and guider's respectively. As the *Learner* learns from the noisy labeled data, the *Guider* model is used to weigh the samples based on uncertainty thus limiting the contributions of unreliable samples to the gradients back propagated. In the following subsection, we provide the analysis of the reason why using the difference between the predictions of the *Guider* and *Learner* models is a good measure for the uncertainty estimation.

### 3.2.2 THEORETICAL ANALYSIS

Our deep generative model allows us to model the correct labels via the observed labels and therefore leads to a plausible way to adjust sample weights during training. We will show in the following theorem that the label correctness is associated with the difference between the *Learner* and *Guider* models during training.

**Theorem 1.** *Suppose that $\theta^*$ is the optimal solution of the optimization problem. $\theta_L$ is the stochastic random variable for SGD (learner) and $\theta_G$ is iterate average (guider) of $\theta_L$, $\theta_G(T) = \mu_T \equiv \frac{1}{T}\int_0^T \theta_L(t)dt$. The distance between $\theta_L$ and $\theta^*$ is consistent with the distance between $\theta_L$ and $\theta_G$ in the sense that*

$$\mathbb{E}\left[\left(\frac{\partial}{\partial \theta_L}\left[\|\theta_L - \theta^*\|^2\right]\right)^\top (\theta_L - \theta_G)\right] > 0 \tag{12}$$

*which holds when the training epoch $T$ satisfies the following relation*

$$T < \frac{1}{\lambda}\frac{\mathrm{Tr}\left\{\mathcal{H}^{-1}\Sigma\right\}}{\mathrm{Tr}\left\{\Sigma\right\}}, \tag{13}$$

*where $\lambda$ is the constant learning rate.*

We can see from Eq.12, during the early training stage, the distance between the learner and the optimal solution increases when the distance between the learner and guider increases and vice versa. This leads to preventing samples with uncertain labels to participate in training and thus reducing the overfitting on potentially wrong labels. The proof of the theorem is given in the Appendix.

## 4 EXPERIMENTS AND RESULTS

Our experiments are designed to evaluate whether UMN is an effective approach to learn a good model with limited annotated training data which include label noise. We compare to popular approaches of supervised learning, semi-supervised learning and robust learning dealing with noisy labels. These methods represent state-of-the-art approaches for model learning with noisy data. Moreover, we also compare the results with the pipeline without using our uncertainty estimation module. Further, we also evaluate the performance of applying the uncertainty estimation module in identifying corrupted labels.

We experiment on a variety of image classification problems with varying degrees of label corruption rates. We use three popular datasets including **MNIST**, **SVHN** and **CIFAR-10**. For a comprehensive evaluation, we set up five different uniform labels corruption rates including $[10\%, 20\%, 30\%, 40\%, 50\%]$.

### 4.1 IMPLEMENTATION DETAILS

In the experiments, supervised deep learning framework means all the label data are directly used for training. For SVHN and CIFAR-10, we experiment a CNN architecture with 13 convolutional neural network as well as the ResNet-101 architecture (He et al., 2015) as baselines for supervised learning approach. To compare with semi-supervised learning approaches, we select two recent popular works. One is named as mean-teacher (MT) proposed by Tarvainen and Valpola (Tarvainen & Valpola, 2017) and the other one is Langevin et al. (Langevin et al., 2018) named Mislabeled-VAE (M-VAE). Langevin et al. (Langevin et al., 2018) only roughly discusses their idea without showing the details of the model architecture and experimental results on popular benchmarks. In this work, we implement the idea in (Langevin et al., 2018) and compare with UMN in all three different datasets. In our experiments, we use the same encoder and decoder architectures for UMN and M-VAE.

In comparison to robust learning approaches, we apply another two recent works including Mentor-Net (Jiang et al., 2017) and Reweight (Ren et al., 2018). For a fair comparison with these works, we follow all the original setting in these papers in our experiments. For each corruption ratio, we ran $5$ experiments by randomly choosing samples to corrupt each time, and report the mean error rate.

As illustrated in Fig. 1, our framework (UMN) is composed of two encoders and two decoders (VAE and C-VAE) along with a classifier. For the experiments in MNIST, We adapt a multi-layer perceptron (MLP) architecture similar to the one proposed in (Kingma et al., 2014), for the VAE, C-VAE and the classifier models. The encoders, decoders and the classifier of our model use a single hidden layer with ReLU activation and an output layer with no activation. For SVHN and CIFAR-10, we apply a 13-layer convolutional neural network (ConvNet) to build our framework. At the first stage of unsupervised learning, $z_a$ is generated via convolution and up-sampling layers. Similar as the first stage, $z_b$ is generated via the encoder framework which is composed of convolution and dense layers. We maintain a moving average of the learner and use the difference between the predictions of learner and guider model to calculate $\epsilon$. The decoder in the semi-supervised learning stage is built with de-convolution and dense layers.

Due to the limitation in space, we detail the model architecture in the Appendix section. We use the Adam optimizer (Kingma & Ba, 2014) with an initial learning rate of 0.001, $\beta_1 = 0.90$ and $\beta_2 = 0.99$. Each experiment has $5$ runs and each run has 350 epochs. In each epoch, UMN uses all the training data including labeled and unlabeled data. For these supervised learning frameworks, we only use the labeled data. Our implementation is based on deep learning framework TensorFlow with a single NVIDIA Tesla P100 GPU.

## 4.2 DATASETS AND RESULTS

### 4.2.1 MNIST

MNIST is a widely used dataset in machine learning community. It includes $60,000$ images with size $28 \times 28$ pixels. In our experiments, we use $50,000$ images for training and $10,000$ images for testing. 100 labeled data samples are used in the experiments. In semi-supervised training, each mini-batch have 5 labeled samples and 95 unlabeled examples. In our compared supervised model, we apply a Multi-Layer Perceptron (MLP) classifier with one hidden layer of $784$ units with ReLU activation. All the results are listed in Table 1.

From the results, we can find that UMN performs much better than other approaches. Besides UMN, M-VAE (Langevin et al., 2018) outperforms other benchmarks. One possible reason may be due to probablistic modeling of the uncertainty. However, M-VAE depends on the pre-defined label corruption ratio which can't be obtained in most real scenarios.

We did not run Reweight (Ren et al., 2018) on the MNIST dataset since at the time of running the experiments an implementation of the approach on noisy MNIST data was not available and our implementation of the approach would not be a fair comparison without having the right parameter settings. However, comparisons are shown on the other datasets as implementation for them was made available by the authors.

### 4.2.2 SVHN

We experiment with the Street View House Numbers (SVHN) datasets. This dataset includes $73,257$ RGB images of $32 \times 32$ resolution belonging to ten different classes. Following the same experimental settings as in (Tarvainen & Valpola, 2017), we use $500$ labeled data samples from SVHN where $50$ data samples per category and we use the rest of the images for unlabeled data in the semi-supervised learning setting. We randomly corrupt the sample labels within each category uniformly with our defined corruption ratio. We also compare with other five different models on SVHN dataset. In our experiment, each batch includes $5\%$ labeled data.

The comparison results are given in Table 1. As listed in this table, we can find that UMN behaves the best in general except for the case when the corruption ratio is $10\%$. However, we do not observe this phenomena in MNIST and CIFAR-10. One of the reasons may be that MT can handle the situation when the noisy data occupy a small portion. The performance of MT model drops significantly as we ingest more mislabeled data. From the result, we can also observe that other two robust learning

Table 1: Comparison of results on different benchmarks. S-1 represents the supervised learning model via 13 conv layers and S-2 represents supervised learning model via ResNet-101. MN and RW indicate the MentorNet and Reweight approach respectively. Error rate percentage % is used as the measurement unit.

| Dataset | Corruption Ratio | Approaches | | | | | | |
|---------|------------------|------|------|------|-------|------|------|-------------|
| | | S-1 | S-2 | MT | M-VAE | MN | RW | **UMN(ours)** |
| MNIST | 10% | 29.4 | 29.5 | 6.6 | 4.3 | 25.6 | - | **2.5 ± 0.14** |
| | 20% | 36.2 | 36.6 | 11.7 | **2.7** | 39.7 | - | 2.8 ± 0.37 |
| | 30% | 41.1 | 43.0 | 14.6 | 7.4 | 45.8 | - | **5.3 ± 3.9** |
| | 40% | 51.3 | 49.0 | 17.9 | 9.9 | 57.7 | - | **5.8 ± 3.8** |
| | 50% | 57.5 | 59.4 | 34.4 | 28.0 | 65.3 | - | **18.0 ± 12.0** |
| SVHN | 10% | 34.9 | 32.0 | **18.0** | 26.1 | 29.1 | 27.3 | 23.9 ± 0.2 |
| | 20% | 46.3 | 46.0 | 45.5 | 35.2 | 42.1 | 39.0 | **30.5 ± 0.3** |
| | 30% | 61.4 | 58.9 | 53.0 | 37.2 | 53.8 | 51.9 | **30.9 ± 0.6** |
| | 40% | 61.8 | 60.5 | 63.9 | 40.1 | 59.1 | 53.7 | **37.1 ± 0.9** |
| | 50% | 65.1 | 63.3 | 65.9 | 48.3 | 62.0 | 57.9 | **43.2 ± 1.4** |
| CIFAR-10 | 10% | 43.2 | 41.3 | 39.2 | 41.2 | 41.2 | 39.4 | **37.8 ± 0.5** |
| | 20% | 46.3 | 44.7 | 42.1 | 43.9 | 42.4 | 41.9 | **39.8 ± 0.8** |
| | 30% | 52.9 | 50.0 | 47.8 | 51.4 | 51.6 | 49.9 | **43.5 ± 0.5** |
| | 40% | 57.1 | 56.9 | 52.2 | 52.1 | 54.1 | 53.9 | **43.5 ± 1.2** |
| | 50% | 63.3 | 61.3 | 65.4 | 56.3 | 57.1 | 58.3 | **51.9 ± 1.7** |

approaches (MentorNet and Reweight) achieve better performance than supervised only but much lower than these Semi-supervised learning methods. One possible reason is that they do not utilize the unlabeled data.

As illustrated in Fig. 3, overfitting is observed in the M-VAE at a higher ratio of corruption. As we discussed previously, one major reason is because of the pre-defined $f(\epsilon)$ which is used as the constant value during the training phase. In contrast, UMN applies the sample-wise uncertainty estimation. It is accurate and more close to the real scenario. Furthermore, the training of UMN also converges faster with better performance.

Moreover, we conduct the experiments and evaluate the model performances using varying numbers of labeled data with the SVHN. We compare the performance for a popular supervised robust learning approach MentorNet(MN) and UMN, for which the test errors shown in Table 2. In UMN, we still utilize all the unlabeled data which are not utilized for labeled data. UMN shows better performance for labeled data less than 1000 labels per class, which suggests the important role of the semi-supervised learning in robust learning and hence justify the purpose of this work.

Table 2: Test errors with Varying Number of Labels for SVHN

| Number of labels | 1000 | 2000 | 3000 | 10000 | 20000 |
|------------------|------|------|------|-------|-------|
| UMN | 23.9 ± 0.2% | 18.1% | 16.3% | 14.8% | 13.2% |
| MentorNet | 30.5 ± 0.3% | 27.9% | 25.1% | 18.3% | 11.2% |

### 4.2.3 CIFAR-10

Next, we run the comparisions on CIFAR-10 dataset (Krizhevsky, 2009). This dataset contains $32 \times 32$ pixels RGB images belonging to ten different classes. To have a fair comparison, we follow the same experimental setting as used in (Tarvainen & Valpola, 2017). We randomly select 100 data samples from each category. In total, $4,000$ labeled data samples are included in the training set. The rest of the training data are used for unlabeled data. The results are summarized in Table 1. From the listed results, we can see that UMN achieves the best performance among all experiments.

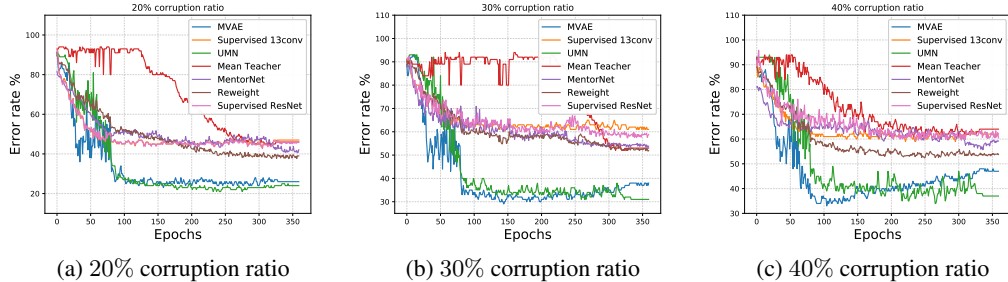

(a) 20% corruption ratio  (b) 30% corruption ratio  (c) 40% corruption ratio

Figure 3: Illustration of error rates' comparisons between UMN and other benchmarks on SVHN during training stage under different corruption ratios ($20\%, 30\%$ and $40\%$). UMN is less prone to overfitting on mislabeled data at higher corruption ratio.

### 4.3 IDENTIFYING SAMPLES WITH CORRUPT LABELS

We have conducted experiments using the MNIST-digit dataset to quantify how well UMN can identify samples whose labels are corrupted. We say, a sample's label is corrupted if there is a disagreement of the predictions between the $Learner$ and the $Guider$ models. That is, if the estimated $\epsilon > 0$ (Eq. 11) for a given sample. Table 3 summarizes the accuracy in identifying samples with corrupt labels. The sample-wise estimate of $\epsilon$ estimated in the final epoch of training is used in the analysis. From the listed results, we can see that UMN has demonstrated very promising potential for filtering out noisy data. It could be easily adopted to other related works as the data pre-processing step. Fig. 4b shows the progression of the uncertainty estimate $\epsilon$ while training on CIFAR-10 dataset with a label corruption rate of $20\%$. We can see that the distribution is bi-modal and that the bi-modal separation increases as the training progresses. i.e., our approach becomes more confident in differentiating samples with corrupt labels from samples with correct labels.

In addition, to better understand the improved performance of UMN in these experiments. We compare the structure of high-dimensional latent feature representations obtained from two VAE components between the model with and the one without using label uncertainty estimation module. We apply t-SNE to the latent feature vector ($z_a$) on CIFAR10 test dataset.

As shown in Fig. 4a, it is clear that the t-SNE maps exhibit different distributions between two architectures. In particular, as indicated by the feature distribution within the black and red boxes, UMN produces a much better separate map between different classes. The used training data had a label corruption rate of $30\%$.

In Fig. 5 we show the distribution of the embeddings for the training samples using different approaches. t-SNE is used for visualizing in 2-dimensions. The training data has 100 labeled samples with $50\%$ label corruption. We can see that the supervised approach clearly overfits on the noisy labels. The semi-supervised approach used is better but one can still see that some of the classes are confused with each other due to label noise, especially when looking at the central region of the plot. However, it can be seen that using the proposed UMN approach results in better intra-class grouping and inter-class separation of training data

Table 3: Performance of identifying samples with corrupt labels.

| corruption ratio | 10% | 20% | 30% | 40% | 50% |
|---|---|---|---|---|---|
| Recall | 1.0 | 0.85 | 0.71 | 0.85 | 0.8 |
| Precision | 0.32 | 0.55 | 0.71 | 0.79 | 0.7 |
| F1 score | 0.48 | 0.67 | 0.71 | 0.82 | 0.75 |

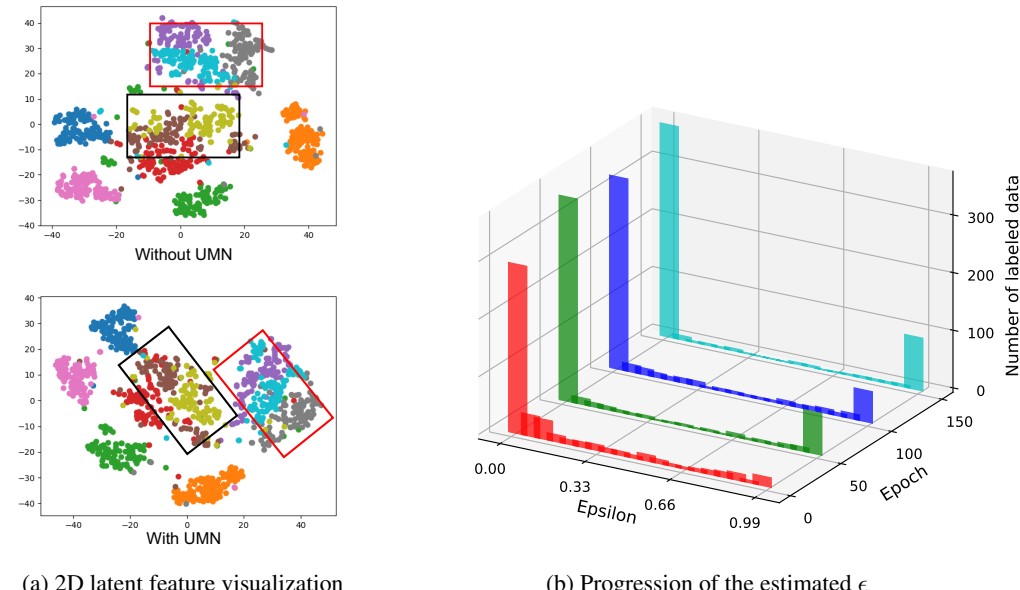

(a) 2D latent feature visualization      (b) Progression of the estimated $\epsilon$

Figure 4: Illustration of latent feature distribution and progression of $\epsilon$. (a) 2D embedding of the learned latent feature $z_a$ (output of encoder in the VAE which serves as the input to the classifier branch) obtained using t-SNE (Maaten & Hinton, 2008) for the MNIST test data. $Left$ and $Right$ plots show distributions obtained $without$ and $with$ the use of uncertainty estimate $\epsilon$, respectively. Different colors in the plots represent different classes (total 10). (b) The distribution visualization of $\epsilon$ evolves along with the time.

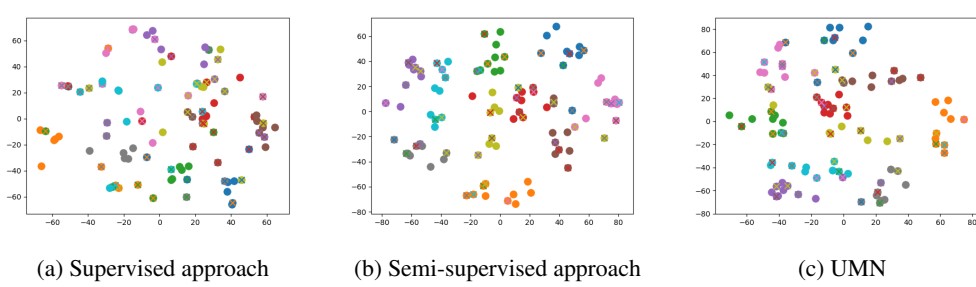

(a) Supervised approach     (b) Semi-supervised approach     (c) UMN

Figure 5: Plots showing the distribution of MNIST training samples in the learned latent space for different approaches. We have used t-SNE to visualize the distribution in 2-dimensions. The different colors indicate the 10 different classes of MNIST. "Circles" are used to represent true labels and "Cross" to represent mislabeled samples. We can see that similarly colored circles (same class) are better grouped by our proposed UMN approach when compared to supervised and semi-supervised approaches. For the experiment, we use the MNIST dataset with 100 labeled samples with 50% label corruption. The network architecture used for UMN is described in section 4.2.1. Using the same architecture, for the supervised experiment we train only the classifier branch of the model, and for the semi-supervised experiment we set the uncertainty estimate to be $\epsilon = 0$.

## 5 CONCLUSION

In this work, we target a new problem of learning on partially labeled data with noise and propose a novel framework (UMN). UMN is a semi-supervised deep generative model dealing with data including scarce and potentially corrupted labels. Moreover, UMN can explicitly estimate the label uncertainty for a given potentially mislabeled data sample. Furthermore, we integrate these frameworks into an end-to-end pipeline. Compared to previous works, we do not need the subset of clean label data or any preknowledge of the corruption ratio. UMN is able to estimate the label correctness based on the uncertainty calculation. Experimental results demonstrate the superiority of UMN over state-of-the-arts for training deep learning models with noisy labels.

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

## 6 APPENDIX

### 6.1 ELBO FOR VAE OF UNCERTAINTY MININING NET (**UMN**)

We start from the Jensen's inequality for the log probability of the observations,

$$\log P(X) \geq \mathbb{E}_{q(z|x)} \log P(X|Z) - \text{KL}(q(z|X) \parallel p(z)) \tag{14}$$

where the observed variables are $X = \{\hat{y}, X\}$ and the latent variables $z$ for the VAE structure we proposed with a stacked semi-supervised learning architecture. The posterior and likelihood factorizes as

$$q(z_a, z_b, y|x, \hat{y}) = q(z_a|x) \, q(y|z_a) \, q(z_b|z_a, y)$$
$$p(x, \hat{y}|z_a, z_b, y) = p(x|z_a) \, p(\hat{y}|y, z_a).$$

The reconstruction loss on r.h.s of Eq. 14 then gives,

$$\sum_{y \in C} \mathbb{E}_{q_\theta(z_a|x)} p_\phi(y|z_a) \left[ \log P(x|z_a) + \log P(\hat{y}|z_a, y) \right]$$
$$= \mathbb{E}_{q_\phi(z_a|x)} \log P(x|z_a) + \mathbb{E}_{q_\phi(z_a|x)} \sum_{y_k \in C} q_\phi(y_k|z_a) \log P_\theta(\hat{y}|z_a, y_k)$$
$$= \mathbb{E}_{q_\phi(z_a|x)} \log P(x|z_a) + \mathbb{E}_{q_\phi(z_a|x)} \sum_{y_k \in C} q_\phi(y_k|z_a) \log P_\theta(\hat{y}|y_k, z_a), \tag{15}$$

where $q_\phi$ and $p_\theta$ are the encoder and decoder respectively for the variational autoencoder and can be parametrized by deep neural network. We note that the second term in the last equality of Eq. 15 is summed over the labeled samples in Semi-supervised learning. The KL divergence in Eq. 14 is given

as

$$- \text{KL}(q(z_a, z_b, y | X, \hat{y}) \parallel p(z_a, z_b, y))$$

$$= \sum_{y \in C} p_\theta(y|z_a) \, \mathbb{E}_{q_\phi(z_a|x)} \, \mathbb{E}_{q_\phi(z_b|z_a,y)} \left[ \log q_\phi(z_a|x, \hat{y}) + \log q_\phi(y|z_a) + \log q_\phi(z_b|z_a, y) \right.$$

$$\left. - \log p(z_a|z_b, y) - \log p(z_b) - \log p(y) \right]$$

$$= - \mathbb{E}_{q_\phi(z_a|x)} \sum_{y \in C} q_\phi(y|z_a) \big( \text{KL}(q(z_b|z_a, y) \parallel p(z_b)) \big)$$

$$- \mathbb{E}_{q_\phi(z_a|x)} \sum_{y \in C} q_\phi(y|z_a) \big( \text{KL}(q_\phi(y|z_a) \parallel p(y)) \big)$$

$$- \mathbb{E}_{q_\phi(z_a|x)} \sum_{y \in C} q_\phi(y|z_a) \mathbb{E}_{q_\phi(z_b|z_a,y)} \big( \log p_\theta(z_a|z_b, y) - \log q_\phi(z_a|x) \big). \tag{16}$$

To summarize, the ELBO for VAE of UMN is

$$- \mathbb{E}_{q_\phi(z_a|x)} \sum_{y \in C} q_\phi(y|z_a) \big( \text{KL}(q(z_b|z_a, y) \parallel p(z_b)) \big) \tag{17}$$

$$- \mathbb{E}_{q_\phi(z_a|x)} \sum_{y \in C} q_\phi(y|z_a) \big( \text{KL}(q_\phi(y|z_a) \parallel p(y)) \big)$$

$$- \mathbb{E}_{q_\phi(z_a|x)} \sum_{y \in C} q_\phi(y|z_a) \mathbb{E}_{q_\phi(z_b|z_a,y)} \big( \log p_\theta(z_a|z_b, y) - \log q_\phi(z_a|x) \big)$$

$$+ \mathbb{E}_{q_\phi(z_a|x)} \log P(x|z_a) + \mathbb{E}_{q_\phi(z_a|x)} \sum_{y_k \in C} q_\phi(y_k|z_a) \log P_\theta(\hat{y}|y_k, z_a), \tag{18}$$

where in the last term, estimations for probabilities of the observed label is given based on the true labels and the corresponding encoded sample $z_a$. The loss given by the ELBO are summed over all the samples except for the last term is summed over the samples with the observed variable $\hat{y}$ available.

## 6.2 ALGORITHM OF UMN

---

**Algorithm 1:** UMN Algorithm

---

**while** *Traning()* **do**

    Sample a minibatch from dataset $S^i$;

    Draw $z_a^i$ from $q_\phi(z_a^i|x^i)$ given $x^i \in S^i$;

    Draw $z_a^i$ from $q_\phi(z_a^i|x^i)$ ;

    **for** *all data $x^i \in S^i$:* **do**

        $y_i \sim q_\phi^b(y_i|z_a^i)$

    **end**

    **for** *all labeled data $x_L^i \in S^i$:* **do**

        $\hat{y}_i \sim p_\theta(\hat{y}_i|y, z_a^i)$ Compute $\epsilon$ based on the uncertainty estimation Eq.(9) via Learner and Guider

    **end**

    Compute the generative process via Eq.(3) ;

    Compute loss function $\mathcal{L}$ via Eq.(5) ;

    $g_\theta = \frac{\partial \mathcal{L}}{\partial \theta}; \quad g_\phi = \frac{\partial \mathcal{L}}{\partial \phi}$ ;

    $\theta = \theta$ - AdamUpdate($\theta$); $\quad \phi = \phi$ - AdamUpdate($\phi$);

    For $\phi$ in classifier $y_i \sim q_\phi^b(y_i|x_i)$, maintain Polyak's moving average for the guider as

    $\phi_G(t) = (1 - \alpha)\phi_L(t - 1) + \alpha * \phi_L(t)$

**end**

---

### 6.3 MODEL ARCHITECTURE OF UMN WITH CONVOLUTIONAL NEURAL NETWORK

The model architecture is listed in the table 4. The plot describe how the classifier and VAE model is setup. The VAE is a stacked autoencoder with **M1** + **M2** Kingma et al. (2014). **M1** is a variational autoencoder with convolution neural networks as the encoder and decoder. **M2** is composed of only fully connected layers.

Table 4: Configuration of the neural networks

| Layers of Classifier | Hyperparameters |
|---|---|
| Input | $32 \times 32$ RGB |
| Translation (shared with MVAE) | Randomly $[\delta x, \delta y] \sim [-2, 2]$ |
| Gaussian noise (shared with MVAE) | $\sigma = 0.15$ |
| Conv2D (shared with MVAE) | filter size: $(3, 3, 128)$, same padding |
| Conv2D (shared with MVAE) | filter size: $(3, 3, 128)$, same padding |
| Conv2D (shared with MVAE) | filter size: $(3, 3, 128)$, same padding |
| Pooling (shared with MVAE) | MaxPool, $(2, 2)$ |
| Dropout(shared with MVAE) | $p = 0.5$ |
| Conv2D (shared with MVAE) | filter size: $(3, 3, 128)$, same padding |
| Conv2D (shared with MVAE) | filter size: $(3, 3, 128)$, same padding |
| Conv2D (shared with MVAE) | filter size: $(3, 3, 128)$, same padding |
| Pooling (shared with MVAE) | MaxPool, $(2, 2)$ |
| Dropout (shared with MVAE) | $p = 0.5$ |
| Conv2D (shared with MVAE) | filter size: $(3, 3, 512)$, valid padding |
| Conv2D | filter size: $(1, 1, 256)$, valid padding |
| Conv2D | filter size: $(1, 1, 128)$, valid padding |
| Pooling | AvgPool, $(6,6)$ |
| Fully Connected + Softmax | $128 \to 10$ |
| **Layers of UMN** | **Hyperparameters** |
| Conv2D | filter size: $(3, 3, 32)$, same padding |
| Flatten | $(8, 8, 32) \to 1152$ |
| Fully Connected | $1152 \to 256$ |
| Fully Connected | $256 \to 1152$ |
| Reshape | $1152 \to (6, 6, 32)$ |
| Conv2D Transpose | $(6, 6, 32) \to (8, 8, 128)$ |
| Up-sampling | |
| Conv2D | filter size: $(3, 3, 128)$, same padding |
| Conv2D | filter size: $(3, 3, 128)$, same padding |
| Conv2D | filter size: $(3, 3, 128)$, same padding |
| Up-sampling | |
| Conv2D (shared with MVAE) | filter size: $(3, 3, 128)$, same padding |
| Conv2D (shared with MVAE) | filter size: $(3, 3, 128)$, same padding |
| Conv2D (shared with MVAE) | filter size: $(3, 3, 128)$, same padding |

### 6.4 UNCERTAINTY ESTIMATION AS DIFFERENCE BETWEEN THE GUIDER AND THE LEARNER

We illustrate this using with a model near the convergence in a convex domain (Mandt et al., 2017), assuming a small label corruption ratio. This suggests that our calculated uncertainty can help focus more on the correct labeled samples.

**Assumption 1** We assume that in the optimization problem, the stationary distribution is constrained to a convex region, where the loss function has a quadratic form

$$\mathcal{L} = \frac{1}{2}\theta^{\top}\mathcal{H}\theta, \tag{19}$$

where $\mathcal{H}$ is the Hessian of the loss surface near minimum and the vector $\theta$ of the model parameters. Without loss of generality, optimal $\theta$ lies on the origin.

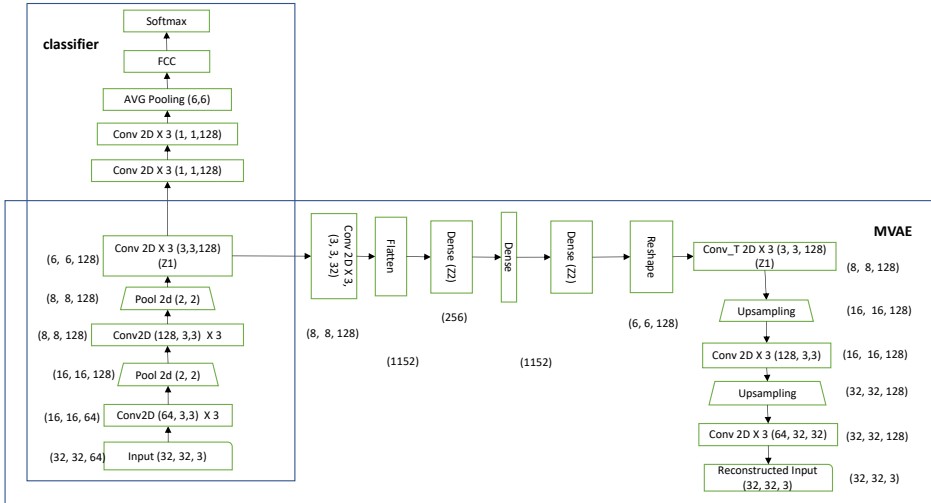

Figure 6: The network structure for Variational Autoencoder for UMN used for SVHN and CIFAR10. The network has two modules. The first one is classifier composed of convolution, pooling, dense and softmax layers. The second one is an $\mathbf{M1} + \mathbf{M2}$ variational autoencoder. The $z1$ of $\mathbf{M1} + \mathbf{M2}$ and the classifier share the same encoder. $z1$ and $z2$ are the two variational autoencoder that encode and decode the raw image.

### 6.4.1 THE CONVERGENCE BEHAVIOR AS STOCHASTIC PROCESS

The convergence behavior of Stochastic Gradient Descent (SGD) can be described by a stochastic differential equation as

$$dθ = -λg(θ)dt + \frac{λ}{\sqrt{S}}B(θ)d\mathbf{W}(t), \tag{20}$$

where $g(θ) \equiv \mathcal{H}θ(t)$ is the gradient for the weights and $S$ is the mini-batch size. $d\mathbf{W}(t)$ represents the Wienner process in the stochastic gradient descent. $B(θ)$ is introduced due to the noise in the stochastic gradient descent process. The covariance of the SGD process $Σ = \mathbb{E}(θθ^\top)$ satisfies $\mathcal{H}Σ + Σ\mathcal{H} = \frac{λ}{S}BB^\top$. $λ$ and $t$ represent the step size and step index respectively.

**Lemma 2.** *The stochastic process for parameter optimization of deep learning problem using only piecewise linear activation function with and without noisy labels can be described by the following Wienner processes,*

$$dθ(t) = -λ\hat{\mathcal{H}}θ(t)dt + \frac{λ}{\sqrt{S}}\hat{B}(θ)d\mathbf{W}(t); \tag{21}$$

$$dθ(t) = -λ\mathcal{H}θ(t)dt + \frac{λ}{\sqrt{S}}B(θ)d\mathbf{W}(t), \tag{22}$$

*where $\mathcal{H} = \hat{\mathcal{H}}$ and $B = \hat{B}$, assuming the learning rate and batch size is the same.*

The proof follows directly from a theorem (Theorem 4) from (Patrini et al., 2017), stating that the curvature of the loss surface (Hessian) is invariant with respect to the noise when the neural network only uses piecewise linear function as its activation function.

### 6.4.2 POLYAK'S AVERAGE FOR THE WEIGHTS OPTIMIZATION

Polyak et. al (Polyak & Juditsky, 1992) proved that the iterate average gives the optimal convergence rate and approximates the optimal by using the average of the iterates online as

$$\theta_L(t+1) = \theta_L(t) - \lambda g_L(\theta_t);$$
$$\mu_{t+1} = \frac{t}{t+1}\mu_t + \frac{1}{t+1}\theta_t \tag{23}$$

and the iterate average after $T$ step is

$$\theta_G(T) = \mu_T \equiv \frac{1}{T}\int_0^T \theta_L(t)dt, \tag{24}$$

where $\theta_L$ and $\theta_G$ indicate the weights of the leaner and guider respectively. Next we prove the theorem 12 that in the presence of the noisy labels, when we update the weights of the learner model, the distance between the learner and the guider suggests the correctness of the labels assuming the invariance of the Hessian of the learner and guider by the lemma.

*Proof.*

$$\mathbb{E}\left[\left(\nabla_{\theta^*}\left[\|\theta_L - \theta^*\|\right]\right)^\top \nabla_{\theta^*}\left[\|\theta_L - \theta_G\|\right]\right]$$
$$=\mathbb{E}\left[\theta_L^\top\theta^{*\top} - \theta^{*\top}\theta_L - \theta_L^\top\theta_G + \theta^{*\top}\theta_G\right]$$
$$=\mathbb{E}[\theta_L^\top\theta_L] - \mathbb{E}[\theta_L^\top\theta_G] - \theta^{*\top}\hat{\theta}^* + \theta^{*\top}\hat{\theta}^*$$
$$=\mathbb{E}[\theta_L^\top\theta_L] - \mathbb{E}[\theta_L^\top\theta_G] \tag{25}$$

In order to show Eq. 13, it is equivalent to show

$$\mathbb{E}[\theta_L^\top\theta_L] < \mathbb{E}[\theta_L^\top\theta_G] \tag{26}$$

We denote

$$\theta_L = \bar{\theta}_S + \hat{\theta}^*; \qquad \theta_G = \bar{\theta}_T + \hat{\theta}^*$$

These are two stochastic optimization process described above for the stochastic gradient descent and the iterate average. $\bar{\theta}_S$ and $\bar{\theta}_T$ can be seen as the deviation from the optimal point. Hence we need to show

$$\mathbb{E}[\bar{\theta}_S^\top\bar{\theta}_S] < \mathbb{E}[\bar{\theta}_S^\top\bar{\theta}_T] \tag{27}$$

The left hand side of Eq. 27 gives

$$\mathbb{E}[\bar{\theta}_S^\top\bar{\theta}_S] = \text{Tr}(\Sigma) \tag{28}$$

In order to evaluate the right hand side of the equation, we recall the Green's function for the Ornstein-Uhlenbeck process,

$$\mathbb{E}(\theta(t)\theta^\top(s)) = \begin{cases} \Sigma e^{-\lambda\mathcal{H}(s-t)}, & \text{if} \quad t < s \\ \Sigma e^{-\lambda\mathcal{H}(t-s)}\Sigma, & \text{if} \quad t \geq s \end{cases} \tag{29}$$

Then, the right hand side of Eq. 27 becomes,

$$\mathbb{E}[\bar{\theta}_S^\top\bar{\theta}_T] = \text{Tr}\left[\frac{1}{T}\int_0^T \mathbb{E}[\bar{\theta}_S^\top(t)\bar{\theta}_S(t')]dt'\right]$$
$$= \text{Tr}\left[\frac{1}{T}\int_0^T e^{-\lambda\mathcal{H}(t-t')}\Sigma dt'\right]$$
$$= \text{Tr}\left[\frac{1}{T}U\Lambda^{-1}(\mathbf{I} - e^{-\lambda T\Lambda})U^T\Sigma\right]$$
$$\approx \text{Tr}\left[\frac{1}{\lambda T}\mathcal{H}^{-1}\Sigma\right] \tag{30}$$

where we assume that the Hessian is full rank and its inverse has an eigen decomposition

$$\mathcal{H}^{-1} = U\Lambda^{-1}U^T \tag{31}$$

When

$$T < \frac{1}{\lambda}\frac{\text{Tr}\left\{\mathcal{H}^{-1}\Sigma\right\}}{\text{Tr}\left\{\Sigma\right\}} \tag{32}$$

The Eq. 13 holds. □

If we make assumption as in Jastrzebski et al. (2017), i.e., covariance of the local minimal is approximated by the Hessian, namely, $\mathcal{H} = \Sigma$. We observe that the last equation in the proof above becomes

$$T > \frac{1}{\lambda}\frac{\text{Tr}\left\{\mathcal{H}^{-1}\Sigma\right\}}{\text{Tr}\left\{\Sigma\right\}} = \frac{1}{\lambda\mathcal{L}}, \tag{33}$$

where $\mathcal{L}$ is the Laplacian. This indicates that our approach favors the loss surface around minimal of a lower mean curvature and hence a narrower minimal.

