# OpenReview forum: "IS THE LABEL TRUSTFUL: TRAINING BETTER DEEP LEARNING MODEL VIA UNCERTAINTY MINING NET"
_ICLR.cc/2020/Conference — Reject_

### Official Review · AnonReviewer1 · 2019-10-22
**Official Blind Review #1**

**Rating:** 3

**Review:**

Update after rebuttal:
The rebuttal addressed a few of my concerns, but there is still a major issue. Namely, UMN is claimed to work on sample-wise label noise, but there are no experiments to support this (note that this is different from non-uniform class-dependent label noise). As fixing this would require large modifications to the paper, I am keeping my score at weak reject.

----------------------------------------------------------------------------------
Summary:
This paper presents a method for training classifiers in the setting of semi-supervised learning with noisy labels. The proposed method, Uncertainty Mining Net, combines the Mean Teacher method of Tarvainen and Valpola with the M-VAE method of Langevin et al. to estimate the trustworthiness of each input-label pair and weigh its contribution to the loss. This is not an obvious use of the Mean Teacher method, and it seems like a nice idea.

The results are good when controlling for architecture, and the setting is important and underexplored, but there are several concerns I have with the paper in its current form and some parts I would like clarified. At present, the paper is borderline, and I will raise my score if these are addressed.

Major points:
The generative process defined in equations 2 and 3 presents a model for input-dependent label noise, but the corruptions in the experiments are conditionally independent of the input given the true label. What is p(y_tilde | y, z_a) supposed to capture when the true noise model in the experiments follows p(y_tilde | y)?

It seems like the proposed approach would have difficulty with non-uniform label noise, but there is no discussion on this. Adding discussion of this would be good.

What is the purpose of z_b? It seems like a redundant variable in the generative process, since it is only used with y to sample z_a, and it seems like z_a could just be sampled from y.

The caption in Figure 1 says, “For simplicity, we omit the optional consistency loss term between the classifiers of θ and θ 0 for unlabeled data in the figure”, but this is never mentioned again. I would find it interesting to see the combined effect of the Mean Teacher consistency loss with the VAE reconstruction loss, since they are distinct approaches to semi-supervised learning. Did you run experiments with the consistency loss?

Minor points:
The writing is full of grammatical errors and typos, including

“we experiment a CNN architecture with 13 convolutional neural network as well as a ResNet-101 architecture”

“the training of UMN also converge faster”

“For the details of the proof, please refer to the appendix section.”
(This quote is from the appendix!)

“eign decomposition”

“for solving learning model”

**Experience Assessment:**

I have published one or two papers in this area.

**Review Assessment: Checking Correctness Of Derivations And Theory:**

I did not assess the derivations or theory.

**Review Assessment: Checking Correctness Of Experiments:**

I assessed the sensibility of the experiments.

**Review Assessment: Thoroughness In Paper Reading:**

I read the paper at least twice and used my best judgement in assessing the paper.

---

> ### Author Response · Authors · 2019-11-15
> **Response to Reviewer 1**
>
> Review#1
> Question1. About the comment from the reviewer “The proposed method, Uncertainty Mining Net, combines the Mean Teacher method of  Tarvainen  and  Valpola  with the M-VAE method of Langevin et al. to estimate the trust worthiness of each input-label pair and weigh its contribution to the loss.”
> Answer1
> Thanks the reviewer for the insightful comment. To clarify our contribution, we formulate a generative model similar to M-VAE to enable sample-wise label noise estimation.  Compared to other approaches of approximating the class-wise noise distribution, UMN is able to estimate the sample-wise label. The sample-wise noise is estimated by labels’ uncertainty of comparing the learning model with a moving average soft label target (guider). This moving average approach is used to provide a guider of the ground truth label and by contrast, the mean teacher is used to enforce temporal consistency. This pipeline enables robust learning without any prior-knowledge of the corruption ration. At the same time, our model also provides the feedback of label uncertainty which can be used as one way of data cleaning.
>
> Question2
> “The generative process defined in Equations 2 and 3 presents a model for input-dependent label noise, but the corruptions in the experiments are conditionally independent of the input given the true label. What is p(y_tilde | y, z_a) supposed to capture when the true noise model in the experiments follows p(y_tilde | y)?
> ”
> Answer2
> We thank the reviewer for raising this question.  We hope the following answers can our answer the reviewer’s concern.  To compare with previous approaches and performance evaluation, we set the class-wise corruption ratio. However, as discussed in the paper, UMN can provide the sample-wise corruption rate estimation which can be used to indicate the mislabeled probability for the given data sample. Therefore, our model is not limited to study the mis-labeled data with uniform noise distribution. Suppose that we have a domain expert and we let him/her examine the label. The domain expert will estimate the label correctness based on the sample --this is exactly the role of the moving average model. Finally, we would like to point out that, if we would like to estimate the class-wise noise distribution, we could also estimate the corruption ratio by summing the noise from each individual.
> Question 3
> “It seems like the proposed approach would have difficulty with non-uniform label noise, but there is no discussion on this. Adding discussion of this would be good.”
> Answer 3.
> Since UMN is able to deal with the sample-wise label noise, the method could handle the situation where the label noise is non-uniform distributions. For further understanding of UMN’s performance when dealing with the non-uniform label noise, we have conducted several experiments. The experimental setting and the results are listed as follows:
> Question 4
> “What is the purpose of  z_b?  What is the purpose of z_b? It seems like a redundant variable in the generative process, since it is only used with y to sample z_a, and it seems like z_a could just be sampled from y.”
> Answer 4
> z_b is the output of the encoder module for a given conditional variation auto-encoder. We adopt the idea from Kingma et. al, 2014.  As indicated in Kingma et. al, 2014, incorporating z_b to the generative process can help improve the performance of semi-supervised learning.
> In our framework, z_a is used for encoding all the data regardless of their labels. z_b is used for generative process conditioned on the correctly estimated label.  The reviewer proposed to use z_b directly -- this is an interesting direction that warrants exploration. Based on our current observation that integrating z_a with z_b works better in UMN.
>
>
> Question5
> The reviewer is wondering “Did you run experiments with the consistency loss?”
>
>
>
> Answer5
> Based on your feedback, we conducted an experiment to evaluate the performance with consistency loss term included.
> Comparisions results:
> Dataset: MNIST-digit (10 classes)
> Corruption rate: 50% (labels for 50% of each class were randomly corrupted)
> The same hyper parameter settings (for UMN) was used for all experiments.
> Error-rate:
> UMN         UMN-Consistency-loss
>   5%                     35%
> The optional consistency loss was not used in our UMN framework. During our development stage, we experimented with it. However, we found that having the consistency loss did not help improve the model performance. The consistency loss would pull the Learner towards the Guider, thus diminishing their differences (thus making it not reliable to estimate label uncertainty). We did not anticipate that our “Guider” would act as a “Mean Teacher” (pun intended).
>
>
> Question6
> The reviewer is commenting on the grammar errors and typos.
>
> Answer6
> Thanks for pointing this out. All the typos have been addressed in the latest version.

---

### Official Review · AnonReviewer2 · 2019-10-22
**Official Blind Review #2**

**Rating:** 1

**Review:**

The paper describes a method for learning in semi-supervised settings with label noise. This is an interesting topic with a relatively scarce literature. The proposed method works by first postulating a generative model for the labelled/unlabeled/label-corrupted data. The model is then fitted using a standard variational lower bound maximization.

The approach is elegant, and appears to work empirically well. Unfortunately, I do not seem to really understand the rationale behind the main novelty (even at an heuristic level) of the paper which is contained in Section 3.2.1, namely Equation (10) which describes the intensity of the label corruption. Why is it a sensible idea? There is only 3 lines of comments after equation 10, which seems a bit short since this is the crux of the paper. Also, the notations of equation (10) are very confusing (to me) since the function f(.) was used before to denote another quantity.
Minor remarks:
=============

1. page4 was a bit difficult to digest at first reading since the author did not describe in the main text he form (i.e. factorization structure) of the variational distribution q.

2. second line of Equation 14 seems to be wrong, while the 1st and last line seems correct -- the authors may want to double check

3. the term "labeled loss term" was not properly defined, which makes the reading a bit difficult even if one eventually gets what the authors mean

4. In equation (6) I do not understand why f(eps) = log[(C-1)(1-eps)/eps]. Why isnt it equal to log[p(hat(y)|y)] ?

5. notations z_a, z_1, etc.. do not seem to be consistent throughout the text

6. it is quite surprising to me that, in the low data-corruption regime, the method appears to be competitive/better with consistency-based method such as the MT approach.

7. naive question: why directly modeling \eps as a function of z_a or (z_a, y), possibly parametrized by a neural network, a bad idea?

In conclusion, it feels like the method has a lot of potential (in views of the numerical simulations) -- if the authors could clarify the exposition, this could be a very good contribution to the field. The method appears to be conceptually simple (i.e. postulate a generative model + fit it by maximizing the ELBO + one trick to estimate \eps), which is a good thing -- what is missing, I think, is a real discussion of why the proposed manner to estimate \epsilon is sensible.

Edit after reading [1]
======================
The proposed generative model is same -- the authors should make this very clear in the paper. Although is is acknowledged: "In this work, we implement the idea in (Langevin et al., 2018)" -- this is only in section "4.1 IMPLEMENTATION DETAILS". After reading (1), it is clear that the novelty of the paper us much less than what I had initially thought. The only difference is in Section 3.2.1, and this Section is far from being satisfying.


[1] "A Deep Generative Model for Semi-Supervised Classification with Noisy Labels", Langevin & al

**Experience Assessment:**

I have read many papers in this area.

**Review Assessment: Checking Correctness Of Derivations And Theory:**

I assessed the sensibility of the derivations and theory.

**Review Assessment: Checking Correctness Of Experiments:**

I carefully checked the experiments.

**Review Assessment: Thoroughness In Paper Reading:**

I read the paper at least twice and used my best judgement in assessing the paper.

---

> ### Author Response · Authors · 2019-11-15
> **Response to Reviewer #2**
>
>
> Question1:  Unfortunately, I do not seem to really understand the rationale behind the main novelty (even at an heuristic level) of the paper which is contained in Section 3.2.1, namely Equation (10) which describes the intensity of the label corruption. Why is it a sensible idea?
> Answer1:
> To clarify, our intent was to show that, under the condition of Equation (12), the Guider model (moving average) leads with its prediction and provides a more reliable indicators for the labels of the individual data sample. When the learner model encounters incorrect labels during training, it deviates from the true optimal and this uncertainty is reflected in eq (10), and eq (8) reduces the "misleading" gradient update due to the incorrect labels. Taken the other way around (i.e. when this misleading gradient is diminished), the Guider model accumulates a “better” gradient and subsequently provides a better label indicator for the learner.
>
> Q2. About  the notations of Equation (10)
> A2 f indicates the model prediction output. Thanks for pointing this out, we will further clarify this part in the revision.
>
> Q3. About  page 4 is difficult to digest.
> A3 The details of how each term is derived from the ELBO are described in the Appendix.  q  is the variational distribution of the hidden variables which we have described in Appendix A.1. q is not fully described in the main text since we followed the convention in the literature of using q to refer to the posterior in the Variational Bayesian. We will provide more details in the main text for better readability.
>  In our work, this q is the distribution of the hidden variable, given the observables. We also describe the p and q distributions as - “where qφ(y|za), qφ(za|x), qφ(zb|za, y), pθ(za|zb, y), pθ(x|za) indicate the classifier, encoder, conditional encoder, conditional decoder and decoder functions respectively”.
>
> Q4. About second line of Equation 14:
> A4 Thanks for pointing this out, the typo has been addressed in the revised version.

---

> > ### Author Response · Authors · 2019-11-15
> > **labeled loss term & some derivations & low noise regime & Parametrization of NN**
> >
> > Q5. About the term “labeled loss term”.
> > Sorry about the confusion. To further clarify, we will include additional introduction right before equation (6).  The sentence after equation (4) will be rephrased as:
> > “where qφ(y|za), qφ(za|x), qφ(zb|za, y), pθ(za|zb, y), pθ(x|za) indicate the classifier, encoder, conditional encoder, conditional decoder and decoder functions respectively. L indicates labeled data. U is unlabeled data and C denotes the set of used classes. The last term in the equation can be treated as the labeled loss term. We will also include the modification in the revised version of the paper.”
> >
> >
> > Q6. About the  equation(6), “I do not understand why f(eps) = log[(C-1)(1-eps)/eps]. Why  isn’t  it equal to log[p(hat(y)|y)] ?”
> > Because we have to sum over y_k ∈ C in the last term of equation(4).
> > For a given observed class - j, using equation(5) to iterate over all y_k, the last term of the equation (4) reduces to equation(6):
> >             q(y_j|z_a)* log[(C-1)(1-eps)/eps] + log(eps/C-1).
> > For simplicity, subscript “j” was not shown in equation (6) and the constant term (log(eps/C-1)) can be ignored as it does not contribute to the gradient backpropagation.
> >
> > Q7. About the comments “low data-corruption regime, the method appears to be competitive with mean teacher model”.
> > As listed in Table 1, mean teacher model performs better than UMN with 10% corruption rate in SVHN dataset. But in most cases in the low data-corruption, UMN behaves better than other semi-supervised learning methods, such as Mean Teacher.  Although most people think that small noise ratio may not hurt the performance of deep learning models, based on our observations, this assumption is not always true when training with limited amount of data. When the labeled data size is small, neural network tends to easily overfit on these mis-labeled data, which could hurt the model performance. That is one of the major motivations of our work.
> >
> > Q8. “naive question: why directly modeling \eps as a function of z_a or (z_a, y), possibly parametrized by a neural network, a bad idea?.”
> > To check the feasibility of using a neural network to estimate the uncertainty (\eps) as a function of the input, using our current objective, we conducted an experiment and results are shown below:
> > Approach A:
> > The \eps estimator was parameterized by 2-layer MLP network with its input as “z_a” concatenated with all possible labels and output is a sigmoid function for \eps for each label.
> > Approach B:
> > The \eps estimator was parameterized by 2-layer MLP network with its input as “z_a concatenated with the output of “classifier” (y-hidden) and output is a sigmoid for \eps.
> > Comparisions results:
> > Dataset: MNIST-digit (10 classes)
> > Corruption rate: 50% (labels for 50% of each class were randomly corrupted)
> > The same hyper parameter settings (for UMN) was used for all experiments.
> > Error-rate:
> > UMN         UMN-Approach-A           UMN-Approach-B
> >  5%                    24.2%                                  27%
> > While the idea of parameterizing \eps using a neural network is interesting, we think it may not be applicable to UMN using the current objective function. It can be derived and shown that with the current objective that the parameterized \eps estimator would be influenced by the parameterized “classifier” - reducing uncertainty for samples that the classifier is confident on and increasing uncertainty for less confident samples. To make it work, it would require an additional supervised term in addition to our current objective function to properly train the network for estimating \eps, which could be explored in future works.

---

> > > ### Author Response · Authors · 2019-11-15
> > > **Generative Process & our contribution**
> > >
> > > Q9. About the  generative process compared to (Langevin et al., 2018)’s paper.
> > > It seems that this is a major concern of the reviewer. We thank the reviewer for drawing this to our attention and allowing us to clarify our contributions.
> > > It is correct that the generative process in Equation (2) resembles (Langevin et al., 2018). However, the key difference is the last component of Equation (3), which describes our generative process of the observed label jointly from the sample and its true label.
> > > After arriving at Equation (3), we did not follow the common way to factorize the distribution p_θ(\hat y| y, z_a) as ref [1]. But instead, we computed it with an estimated uncertainty term which ultimately enables evaluation of the probability of individual sample’s label noise. We provide proof in the Appendix for why our way of estimating label uncertainty (\eps) is a sound approach.
> > > By contrast, Langevin et al., 2018 use pre-defined constant values as the class-wise corruption rates in their generative process which leads to gradient backpropagation being modulated class-wise. They use the same corruption rate for all data samples within a category. However, it is not practical to obtain the corruption rate of mis-labeled data as pre-knowledge.  And using a constant value as the corruption rate for all samples of a given class is not precise during the training, e.g., the ratio of mis-labeled and correctly-labeled data in different training batches may be quite different. Which is why we were motivated to model the label uncertainty per sample instead of per class in the generative process.
> > > Reference:
> > > [1] Bo Han, Jiangchao Yao, Gang Niu, Mingyuan Zhou, Ivor Tsang, Ya Zhang, and Masashi Sugiyama. Masking: A new perspective of noisy supervision. arXiv preprint arXiv:1805.08193, 2018.

---

### Official Review · AnonReviewer3 · 2019-10-24
**Official Blind Review #3**

**Rating:** 6

**Review:**

In this paper, the authors proposed a novel framework, Uncertainty Mining Net (UMN), to address the problem of learning on limited labeled data with label noise. First, UMN applied the unsupervised variational autoencoder to learn a more representative latent feature representation by involving large amounts of unlabeled data. Second, in the training process, ELBO integrates information from all the training data and sample-wise uncertainty estimated from the predictions of the updating model and its exponential moving average is incorporated to enable the learning process to focus on the data with reliable labels. Experimental results show that UMN outpuroms several state-of-the-art methods on multiple benchmark datasets.

UMN will be helpful for a lot of real-world products since annotation is expensive and annotation quality is a regular concern. However, it is subjectively uncertain to define whether it is a large or small dataset without considering application context and model complexity. It would be better if the authors can further provide the error rate changes under different number of labeled training data. Such analysis would provide suggestions regarding when it is necessary to implement UMN. On the other hand, it would be nice to explore how tolerant/sensitive the UMN is to the corruption rate to every single class, especially for a multi-classification problem. In the real world, the data within the same class might be labeled incorrectly more than correctly. It would be better to investigate if UMN is able to identify such situation and correct the labels accordingly.

**Experience Assessment:**

I have read many papers in this area.

**Review Assessment: Checking Correctness Of Derivations And Theory:**

I assessed the sensibility of the derivations and theory.

**Review Assessment: Checking Correctness Of Experiments:**

I assessed the sensibility of the experiments.

**Review Assessment: Thoroughness In Paper Reading:**

I read the paper at least twice and used my best judgement in assessing the paper.

---

> ### Author Response · Authors · 2019-11-15
> **Response to Review #3**
>
> For Review #3.
> Question 1: It would be better if the authors can further provide the error rate changes under different number of labeled training data. Such analysis would provide suggestions regarding when it is necessary to implement UMN.
> Answer 1:
> Thanks for the insightful comment. Following your suggestions, we conduct several experiments accordingly  to evaluate the sensitivity of UMN to different number of labeled training data.
> We setup three different experiments where the training set includes 200, 300 and 500 labeled data samples respectively, the rest of the data is used as the unlabeled data for the semi-supervised learning. For comparison, we apply the same experimental setting to one of the latest supervised robust learning works –MentorNet.
> The followings are the error rates for different experimental settings. There is no pre-training in the experiments.
>
> SVHN dataset,
> For 20% of corruption ratio:
>
>                                        MentorNet .                  UMN.
> 100 samples/class:       30.5%                     23.9%
> 200 samples/class:       27.9%                     18.1%
> 300 samples/class:       25.1%                     16.3%
> 1000 samples/class:    18.3%                      14.8%
> 2000 samples/class:    11.2%                      13.2%
> From the results, we can see UMN still works with larger label training data and even achieve better performance compared to MentorNet under most situations. This suggests that the unlabeled data is important for assisting training with noisy labeled data which justify the purpose of our work. We thank again the reviewer for this import piece of advice.
>
>
>
> Question 2. In the real world, the data within the same class might be labeled incorrectly more than correctly. It would be better to investigate if UMN is able to identify such situation and correct the labels accordingly.
>
>
>
>
>
> Answer2:
>
> To investigate the performance with larger corruption rates, we have evaluated the model performance on MNIST/SVHN with corruption rate from 60% to 90%. The performance is listed as follows. From the results, we can tell UMN still can achieve reasonable performance under the situations when there are more mis-labeled data than the correctly labeled data.
>
>
>
>                             MentorNet,           UMN,
> 60%                   67.1%               49.1% (another 50 epochs to run)
> 70%:                 70.0%               53.1%
> 80%:                 71.2%               54.5% (another 10 epochs to run)
> 90%:                  ---                     ----
> Thanks for these valuable comments, we have included this ablation study in the paper and we will try to add these additional ablation study results to the paper.

---

### Decision · Program_Chairs · 2019-12-19

**Decision:**

Reject

**Comment:**

The paper presents an interesting idea but all reviewers pointed out problems with the writing (eg clarity of the motivation) and with the motivation of the experiments and link to the contest. The rebuttal helped, but it is clear that the paper requires more work before being acceptable to ICLR.